# A Compilation of Snow Cover Datasets for Svalbard: A Multi-Sensor, Multi-Model Study

Hannah Vickers [1,*], Eirik Malnes [1], Ward J. J. van Pelt [2], Veijo A. Pohjola [2], Mari Anne Killie [3], Tuomo Saloranta [4] and Stein Rune Karlsen [1]

1. NORCE Norwegian Research Centre AS, P.O. Box 6434, NO-9294 Tromsø, Norway; eima@norceresearch.no (E.M.); skar@norceresearch.no (S.R.K.)
2. Department of Earth Sciences, Uppsala University, 75105 Uppsala, Sweden; ward.van.pelt@geo.uu.se (W.J.J.v.P.); Veijo.Pohjola@geo.uu.se (V.A.P.)
3. Norwegian Meteorological Institute, P.O. Box 43, NO-0313 Oslo, Norway; mariak@met.no
4. Hydrology Department, Norwegian Water Resources and Energy Directorate, P.O. Box 5091, NO-0301 Oslo, Norway; tus@nve.no
* Correspondence: havi@norceresearch.no

**Abstract:** Reliable and accurate mapping of snow cover are essential in applications such as water resource management, hazard forecasting, calibration and validation of hydrological models and climate impact assessments. Optical remote sensing has been utilized as a tool for snow cover monitoring over the last several decades. However, consistent long-term monitoring of snow cover can be challenging due to differences in spatial resolution and retrieval algorithms of the different generations of satellite-based sensors. Snow models represent a complementary tool to remote sensing for snow cover monitoring, being able to fill in temporal and spatial data gaps where a lack of observations exist. This study utilized three optical remote sensing datasets and two snow models with overlapping periods of data coverage to investigate the similarities and discrepancies in snow cover estimates over Nordenskiöld Land in central Svalbard. High-resolution Sentinel-2 observations were utilized to calibrate a 20-year MODIS snow cover dataset that was subsequently used to correct snow cover fraction estimates made by the lower resolution AVHRR instrument and snow model datasets. A consistent overestimation of snow cover fraction by the lower resolution datasets was found, as well as estimates of the first snow-free day (FSFD) that were, on average, 10–15 days later when compared with the baseline MODIS estimates. Correction of the AVHRR time series produced a significantly slower decadal change in the land-averaged FSFD, indicating that caution should be exercised when interpreting climate-related trends from earlier lower resolution observations. Substantial differences in the dynamic characteristics of snow cover in early autumn were also present between the remote sensing and snow model datasets, which need to be investigated separately. This work demonstrates that the consistency of earlier low spatial resolution snow cover datasets can be improved by using current-day higher resolution datasets.

**Keywords:** polar regions; snow cover; remote sensing; snow modelling; MODIS; Sentinel-2

## 1. Introduction

Snow cover is a crucial component of the climate system, with its high albedo allowing up to 90% of incoming solar radiation to be reflected. Snow is also an important insulator, and in cold climates such as those found in the high latitude regions, it protects underlying soil and vegetation from frost damage. However, past and present changes in the global climate have been producing pronounced effects in the polar regions, as increasing temperatures lead to loss of snow, glacier and sea ice cover which in turn reduce the surface albedo and increase absorption of solar radiation, producing even greater warming [1]. The Svalbard archipelago, located in the High Arctic, is heavily glaciated and glaciers alone make up 57% of the total land area of Svalbard [2]. However, as a result of a warming climate,

the spatiotemporal characteristics of seasonal snow cover on Svalbard have undergone significant changes in the past two decades, with large parts of the archipelago exhibiting trends of earlier spring snowmelt and disappearance [3,4]. Projected changes in climate, as outlined in a recent report on the future climate of Svalbard, indicate that by 2100, increases of 3–4 °C and 6–8 °C in the mean annual temperature can be expected for the west coast and northeastern regions respectively, compared with the 1961–1990 average [5]. Such marked warming will inevitably lead to significant impacts across the cryosphere, hydrosphere and biosphere, as changes in snow and ice cover affect timing and intensity of surface runoff and water storage and availability. On average, predictions for Svalbard indicate more than doubling of the snow-free season length and total runoff from glaciers and seasonal snow between 1957–2018 and 2019–60 for a RCP4.5 emission scenario [6]. Snowpack water content is an important component of the hydrological cycle; therefore, snow cover mapping is useful in both assessing water resources and for calibration of hydrological models [7–9] through data assimilation [10]. Up-to-date, detailed information on snow cover and conditions is also an important element for forecasting of natural hazards such as avalanches, slush flows and snowmelt floods, all of which may occur more frequently in a warming climate. Knowledge of snow water equivalent (SWE) in mountain catchments is also crucial to the hydropower industry, especially for management of seasonal water resources. Operational daily maps of simulated snow conditions have already existed for 15 years for mainland Norway. However, there is an absence of detailed, spatiotemporal information of snow conditions on Svalbard, despite the obvious relevance and need for such information in, for example, natural hazard forecasting on Svalbard and planning of outdoor and tourism activities.

The evolution of snow parameters can be simulated continuously in space and time through utilization of snow models. These require a surface meteorological forcing, which is either obtained from output of regional climate/numerical weather prediction models or reanalysis datasets for large-scale modelling. The evolution of the seasonal snowpack over land in Svalbard is dominated by snow accumulation during autumn and winter and subsequent melting during late spring and summer. Snow accumulation and spring maximum snow depth is mostly determined by cumulative precipitation in the form of snow during autumn and winter, while snowmelt depends on land-atmosphere interactions that can be estimated using simple melt-air temperature relationships such as the positive-degree day model, or more sophisticated models that solve the surface energy balance. Snow models therefore represent a valuable tool for filling spatial and temporal gaps in observational datasets. Moreover, they can simulate snow over longer time-periods and larger spatial domains than observational datasets. Essential to snow model calibration and validation is the use of in situ and/or remote sensing snow products, estimates of SWE, snow depth, density, temperature and water content. However, measuring snow parameters traditionally by means of in situ observations provide only point measurements and is limited in spatial coverage. Furthermore, the installation and maintenance of networks of meteorological instruments is often challenging in high mountain and remote terrain environments.

Remote sensing of snow cover provides a means of observing snow cover over large spatial areas that cannot be fulfilled by in situ observations alone and has been well-reviewed in recent years [11,12]. Optical sensors make detection of snow possible by utilizing the reflectance characteristics of snow at different wavelengths. Snow is distinguishable from other types of surface cover due to its high reflectance properties at visible wavelengths, low reflectance in the near infrared band and shortwave infrared wavelengths [13]. Several generations of optical sensors have now been acquiring data globally for several decades; the Moderate Resolution Imaging Spectroradiometer (MODIS) onboard the Terra and Aqua satellites has been acquiring optical images since 2000, from which the Normalized Difference Snow Index (NDSI) can be derived [14] as well as fractional snow cover [15]. Recently, a 20-year MODIS snow cover fraction (SCF) dataset for Svalbard based on the NASA MOD10A1-product [16] at 500 m spatial resolution has been

produced and investigated [4]. Other spaceborne optical sensors include the Advanced Very High-Resolution Radiometer (AVHRR) instrument, which has flown onboard polar orbiting satellites since the late 1970s and provides observations for monitoring snow cover extent (SCE). The instrument has approximately 1 km spatial resolution, but only data at a reduced effective resolution of approximately 4 km is permanently archived and available with global area coverage (GAC). Meanwhile, newer more sophisticated optical sensors on board the Sentinel-2 A and B satellites have been delivering data over Svalbard since 2016 at a nominal 10 m pixel spacing. Since the launch of the Sentinel-2B satellite in 2017, daily coverage of Svalbard has also been possible, therefore providing unrivalled opportunities to study snow cover changes in Svalbard at high temporal and spatial resolution. A recent study has begun to address the similarities and differences in fractional snow cover retrievals using three optical sensors at different resolution and extracted using retrieval algorithms for a study site in northwest Svalbard [17]. The remote sensing observations were further validated by very high-resolution terrestrial photography. Even though the study area and time period were limited in extent, the results nevertheless indicate that there exist discrepancies when comparing observations from lower and higher resolution sensors, as well as the methods used to retrieve them. Furthermore, compiling long term climate records and linking observations to climatic variations by combining data from different sensors that cover different time periods inevitably becomes challenging due to mixed-pixel problems creating biases in fractional snow cover estimates when aggregated over large areas [18].

Despite the existence of multiple optical satellite datasets and snow model datasets, there is an obvious lack of continuity and consistency with respect to spatial resolution and periods of data coverage. Moreover, few attempts have been made that demonstrate how current-day, high resolution remote sensing datasets can be used to reconstruct and upscale snow cover observations of the earlier, low resolution datasets that often provide long time periods of data coverage. This is especially true for the high latitude Svalbard archipelago, where changes in seasonal snow are occurring faster than snow-covered areas at lower latitudes. Furthermore, studies of snow cover to date have often only utilized either remote sensing or modelling and there is therefore a need for large scale comparisons between different resolution sensors and models as well as evaluating the relative strengths and weaknesses of each dataset. Therefore, the objective of this study is to demonstrate the similarities and differences of snow cover observations made using remote sensing datasets and snow models, and how these differences can affect the extraction of derived parameters linked to the dynamical processes such as snow melt and disappearance. We examine snow cover fraction derived using the AVHRR dataset at 4 km resolution and Sentinel-2 observations produced at 20 m resolution and how they compare with the recently published MODIS SCF dataset for Svalbard at 500 m resolution [4]. In addition, two independent snow models developed by the University of Uppsala (Energy balance—snow and firn model; EBFM) and the Norwegian Water Resources and Energy Directorate (seNorge) that provide estimates of SWE and fractional snow-covered area, are used to derive snow cover extent maps. These are compared with the snow cover extent derived from binarization of the MODIS SCF maps to examine the temporal and spatial differences in snow cover. The area of study for this work is defined by the overlapping area common to all datasets available. For this work, we have therefore carried out the data analysis for the Nordenskiöld Land region in the central part of Svalbard.

An overview of the study area will be presented in Section 2 together with a description of the remote sensing and snow model datasets and an outline of the data processing and analysis methods used to perform the comparisons between the datasets. The results of the comparisons as well as a quantitative evaluation of the consistency between the datasets are presented in Section 3. In Section 4, a discussion of the main results is made in the context of current knowledge and earlier studies of relevance. A summary of the primary findings of this study, as well as suggestions for further work is given in the conclusion, in Section 5.

## 2. Materials and Methods

### 2.1. Study Area

The Svalbard archipelago is a group consisting of nine islands, located approximately halfway between the northernmost point of the Norwegian mainland and the North Pole. The archipelago covers a total area of 61,000 km$^2$ of which 60% is glaciated and the remaining area is covered by barren rock and vegetation. The archipelago is spread over latitudes in the range 74–81°N and longitudes ranging from 10–35°E. Both midnight sun and polar night are therefore present for a large proportion of the year. Nordenskiöld Land is situated approximately in the center of the archipelago in terms of latitude and it is in this region the administrative center Longyearbyen is located. The climate of this area is affected by the West Spitsbergen Current, which brings warm salty water from the Atlantic Ocean northwards, producing a milder climate than experienced at similar latitudes elsewhere. Meteorological data recorded at Svalbard Airport, close to Longyearbyen, show that the mean annual temperature in the period of study (2000–2019) ranges from $-6.1$ °C (2003) to $0$ °C (2016), while annual precipitation ranges from a minimum of 142.1 mm (2005) up to 310 mm (2016).

Figure 1 illustrates the area of study chosen for this work. This study area corresponds to about one and a half Sentinel-2 tiles for which NDSI data have been processed and is also the area that overlaps with all datasets that have been compared.

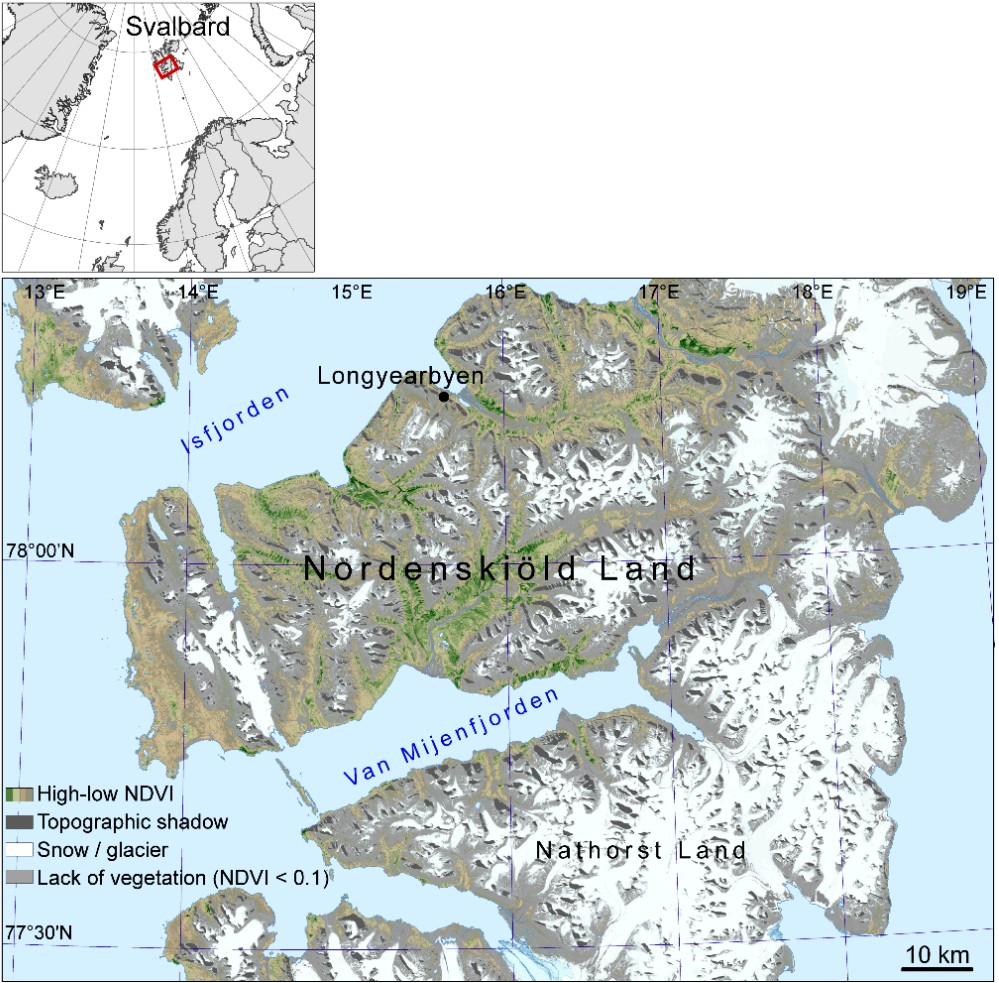

**Figure 1.** Study area of central Svalbard. Sentinel-2 based map from 1 August (mean 2016–2019) with colors extracted from NDSI and NDVI (Normalized Difference Vegetation Index) values.

## 2.2. MODIS

The MODIS instrument on board the Terra and Aqua satellites provides multispectral imagery at visible-shortwave infrared (VSWIR) wavelengths, with a daily revisit period and spatial resolution of approximately 500 m for all VSWIR bands. This study uses the NASA MODIS/Terra Snow Cover Daily L3 Global 500 m Grid, Version 6 (MOD10A1) dataset to derive NDSI as a daily product [16], which is calculated using the spectral band 4 (visible light) and band 6 (short wave infrared) through the relation,

$$\text{NDSI} = ((\text{band4} - \text{band6})/(\text{band4} + \text{band6})) \tag{1}$$

Snow cover fraction, expressed as a percentage, is then estimated using a universal approach [15] defined by the regression formula,

$$\text{SCF} = (0.06 + 1.21\,\text{NDSI}) \times 100 \tag{2}$$

During the polar night when there is no optical data coverage, SCF is set to 100 %. As such, the MODIS SCF estimates are produced only for the period 1 March–1 November for the entire periglacial landscape in Svalbard as a temporally interpolated product at daily intervals and 500 m resolution. In addition, cloud cover can often contaminate MODIS data, which is detected and masked out. There are a number of reconstruction approaches that have been developed to produce daily cloud-free MODIS snow products [19,20]. However, in this study we have implemented a temporal interpolation technique to obtain cloud-free maps for the Svalbard archipelago, which has been outlined in earlier work [4]. The reader is referred to this work for a more detailed description of the method. Since MODIS has moderate spatial resolution and excellent temporal overlap with both older and newer available satellite and modelled data products being used in this study, the MODIS dataset is used throughout this study as the baseline for comparisons.

## 2.3. AVHRR

The AVHRR Global Area Coverage (GAC) data are used to produce a fundamental climate data record (FCDR) for radiances and brightness temperatures. This dataset has been made available by the EUMETSAT Climate Monitoring Satellite Application Facility (CM SAF). The current release, CLARA-A2, covers 1982–2015 [21]. A time series of daily snow cover maps covering the Svalbard archipelago at 4 km grid spacing was derived from the CLARA-A2 FCDR using the probabilistic snow cover algorithm provided by MET Norway. This snow cover algorithm uses a set of instrument channel combinations and statistical coefficients, the latter of which are derived from prior knowledge of the typical behavior of the surface classes across the spectrum. Cloud-free pixels from the AVHRR GAC swath products are averaged and gridded to produce daily maps of average snow probability, to which a threshold of 50% is applied to derive a binary snow cover extent product. Data are provided only for the period between March 1 and September 30 each year due to no data availability during the polar night. This results in one less month with data to compare with MODIS for the overlapping period. In addition, temporal gap filling was applied to achieve daily cloud-free mosaics using information from cloud-free pixels up to 9 days forward or backward in time. The product also indicates the age of the reference image used to make the cloud cover corrections and is the product used for comparisons with MODIS data. Furthermore, a vegetation map for Svalbard produced using Landsat data [22] is used to mask out glaciers and water bodies in the AVHRR dataset, as was done to produce the MODIS SCF dataset.

## 2.4. Sentinel-2 (S2)

In this study we use all the available Sentinel-2 data for the period 15th April to 15th September, each year from 2016 to 2019. From 1 July 2017 data from Sentinel-2B is acquired in addition to Sentinel-2A. From this point onwards, data are available from both twin satellites on most of the days for the study area. Due to the low solar elevation angle, both

early and late in the snow season (solar zenith angles higher than 70°) the Sentinel-2 Level 2A (bottom-of-atmosphere) data are not reliable, and we only used scene classification data from Level 2A as reference data and rely on Level 1C data (top-of-atmosphere) in the snow mapping.

For cloud detection we examined the cloud probability in Level 2A (Sen2Cor processor) and "s2cloudless" machine-learning based algorithm [23]. In addition, we developed our own cloud detection algorithms from multi-spectral values and multi-temporal tests. However, none of the methods work well for sparsely vegetated areas (bright surfaces), which is very common in the study area. For thin semi-transparent clouds, and for cloud shadows the algorithms did not show sufficient accuracies. To detect clouds, we performed a visual inspection in the visual and SWIR bands, and masked out cloud free areas, only using the cloud masks as a reference. One exception was the cirrus clouds, which could be accurately detected with band 10. However, cirrus clouds only appear in a few of the images. This time-consuming method for cloud detection based on visual inspection, ensures fewer errors, but not all cloud free data are included. Some of the images had many small cumulus clouds which were too time-consuming to mask out and so were not used.

NDSI values were calculated (Equation (1)) for the cloud-free pixels and interpolated to produce daily NDSI maps at 20 m resolution. This was achieved by performing linear interpolation and smoothing with a Savitzky–Golay filter. In this study we have primarily focused on retrieving estimates of the land-averaged snow cover fraction by applying a fixed threshold of 0.4 [16,24] to the Sentinel-2 NDSI which produces a binary (snow/no snow) snow cover extent map. We have chosen to utilize a thresholding method since regression coefficients have not been thoroughly tested and validated on Sentinel-2 data yet, as has previously been done for the MODIS datasets.

### 2.5. SeNorge

The seNorge snow model [25] requires 3-hourly or daily mean air temperature and a sum of precipitation as input forcing. Solid precipitation is defined as precipitation occurring at an air temperature ≤0.5 °C. Snow and ice melt are calculated using the extended degree-day model including air temperature and solar radiation terms. The two parameters of the melt algorithm have subsequently been estimated based on 3356 quality controlled daily melt rates observed by the Norwegian snow pillow network [26]. The sub-grid snow distribution algorithm in the model [25] assumes a uniform probability distribution of snow amounts within the grid cells. In addition, an even layer of new snow can form on top of the uniformly distributed "old" snowpack and snow-covered fraction is then set to 1. The main effect of the sub-grid snow distribution is to reduce the grid cell average melting rates towards the late melt season rates when significant areas of bare ground are present in the grid. The 3-hourly input data are aggregated from the hourly meteorological forcing data obtained and downscaled from the AROME Arctic numerical weather prediction model (NWP). Input precipitation in the current model application is scaled by a factor 0.75, based on initial evaluation of the first model results. Model parameter values are set to the same values as those in the application for mainland Norway, except the spatial snow distribution parameter CF is increased from the default value of 0.5 to 0.85, giving larger variance for sub-grid snow distribution. The model application for Svalbard starts at bare ground initial conditions in September 2012. Following this, snow/firn older than 1 year is removed from the model's snow store on September 1 each year. The two first snow seasons may therefore be considered as a model spin-up period at higher elevation areas with perennial snow. SCF estimates for this study are provided at 3-h intervals daily for the years 2013–2019. This study has utilized the seNorge SCF product corresponding to 1200 UTC.

### 2.6. EBFM

The coupled energy balance—snow and firn model (EBFM) [27] has been used to study the long-term climatic mass balance of glaciers [28], as well as seasonal snow conditions

and runoff on glaciers and land since 1957. EBFM solves the surface energy balance to calculate surface melt and temperature, which provides upper boundary conditions for a subsurface model, simulating the multi-layer evolution of snow density, temperature and water content [27]. In Van Pelt et al. (2019) [28], the model was forced by downscaled meteorological fields of precipitation, air temperature, relative humidity, wind speed and air pressure from the High-Resolution Limited Area Model (HIRLAM) [29]. For calibration and validation of the model and meteorological downscaling, in situ on-glacier measurements of weather conditions, stake mass balance and subsurface density were utilized; no calibration or validation was performed for snow conditions in non-glacier terrain, potentially deteriorating performance in these areas. For more details about the methods and dataset, the reader may refer to [28] and references therein. From this large dataset, SWE is extracted across Svalbard at $1 \times 1$ km spatial resolution and daily temporal resolution for 2000–2019. The dataset overlaps for a longer part of the MODIS period, as well as with the seNorge snow model dataset.

### 2.7. Processing of the Datasets

For all datasets being compared only the period for which MODIS data are both available and overlap with the other datasets are used. That is to say, the period where SCF is assumed to be 100% (November–February) is excluded from the study. Table 1 summarizes the spatial resolution, period of data availability and actual time period that has been compared with MODIS.

**Table 1.** Summary of the datasets used in the study, together with the spatial resolution, time period used and type of snow cover product provided.

| Dataset | Data Cover | Years | Resolution | Snow Product |
|---|---|---|---|---|
| **MODIS** | 1 March–31 October | 2000–2019 | 500 m | Snow cover fraction |
| **AVHRR** | 1 March–30 September | 2000–2015 | 4 km | Snow cover extent |
| **Sentinel-2** | 15 April–15 September | 2016–2019 | 20 m | Normalized Difference Snow Index |
| **seNorge** | 1 January–31 December | 2013–2019 | 1 km | Snow cover fraction |
| **EBFM** | 1 January–31 December | 2000–2019 | 1 km | Snow water equivalent |

In Figure 2, a simple flowchart illustrates the workflow and extraction of the parameters of interest for the area studied. For all datasets being compared with MODIS, a resampling was performed in order to reproduce the datasets with a common grid and identical coverage area. The land-averaged snow cover fraction (SCF) was calculated for the study area using either the binary or fractional snow cover products. For all snow products, glaciers and water bodies were excluded from the spatial averaging. SCF was computed for only the periods with overlapping data coverage.

As described in Section 2.4, a fixed threshold of 0.4 was applied to the Sentinel-2 NDSI maps to extract binary snow cover extent maps. For the EBFM SWE datasets, an optimum threshold was determined using the MODIS SCF as a reference dataset. Ten different thresholds on SWE ranging from 0.001 to 0.01 m were used to first obtain a binary snow cover maps and subsequently to calculate the corresponding land-averaged snow cover fraction time series for each threshold. The squared difference of the EBFM and MODIS-derived SCF time series was calculated and summed over the full year, for each year in the dataset. The threshold producing land-averaged SCF time series that gave the smallest squared-sum was identified as the best threshold for that year. Since the optimum threshold

varied from year to year, the median value of 0.003 m for all 20 years was taken as a fixed threshold to be applied to the SWE data.

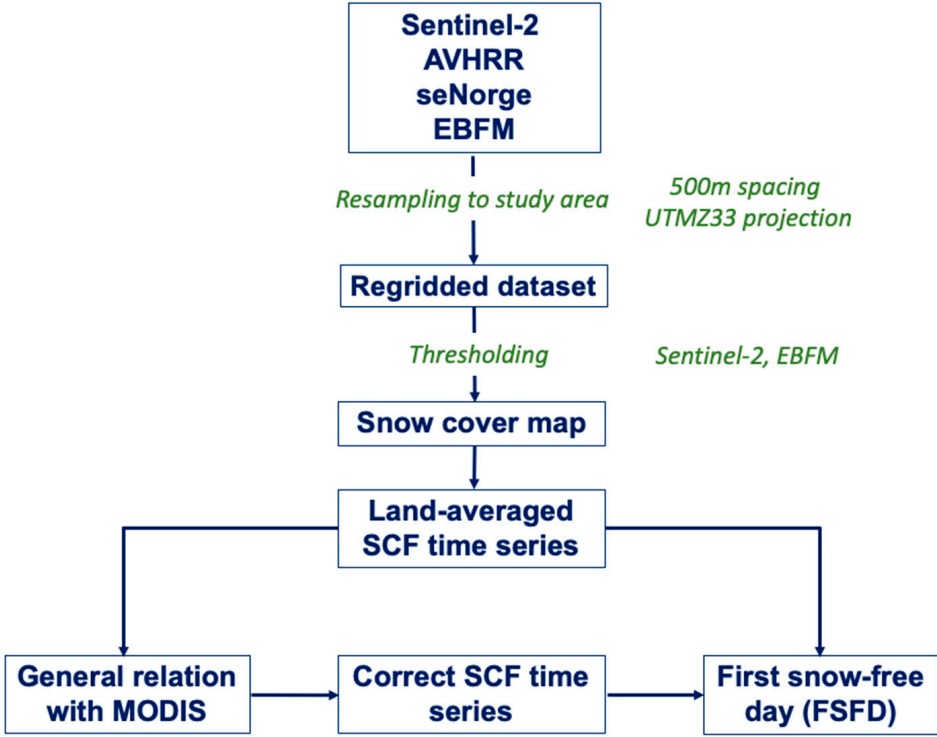

**Figure 2.** Flow diagram illustrating the data processing and parameters extracted from the datasets.

For all datasets, a general relationship was obtained with the MODIS time series by making a cubic spline fit to each pair of datasets (MODIS-AVHRR, MODIS-S2, MODIS-seNorge, MODIS-EBFM). These relationships were further utilized to demonstrate how simple corrections to the datasets can be used to obtain better consistency with the baseline MODIS dataset. This procedure is described in greater detail in Section 3. Lastly, an estimate of the first snow-free day was made using the uncorrected and corrected land-averaged SCF time series in order to study the effect of the corrections on the timing of snow disappearance deduced from the datasets.

## 3. Results

In this section, we present the results of the data processing and analysis outlined in Section 2. This section is divided into five subsections, which describe the different aspects of the data comparisons made. In Section 3.1, a description of the general relationship between the datasets is made, while in Section 3.2, we present a more specific comparisons of the geographical and altitudinal differences between the datasets. Section 3.3 is dedicated to the results of the normalization, or correction of the snow cover time series using the results of Section 3.1, and in Section 3.4, we quantify the effect of the corrections on derived estimates of first snow-free day, compared with the original time series. A quantitative evaluation of the differences between the datasets, before and after corrections is made in Section 3.5.

### 3.1. General Relationship between the Datasets

In this section, a comparison of the snow cover fraction time series is made for each dataset, with respect to the MODIS SCF time series. These comparisons are made using the SCF which is obtained by averaging the snow cover products over the study area. Figure 3 shows the time series of the land-averaged snow cover fraction for all datasets used in this study. There is some overlap between datasets: for example, between MODIS, AVHRR and

the EBFM dataset, from 2000–2019, and from 2013–2015, there is overlap between MODIS, AVHRR, EBFM and the seNorge datasets. In the final four years of the period (2016–2019), there is overlap between MODIS, seNorge, EBFM and Sentinel-2. There is good agreement in the SCF minimum values for MODIS, AVHRR and EBFM for the first five years of the period, though with AVHRR exhibiting greater and more frequent fluctuations in SCF during the summer minimum compared with MODIS and EBFM. From 2006 onwards the fluctuations in SCF derived from AVHRR during the minimum period become more pronounced; moreover, the SCF values during this period also tend to be some tens of percent greater compared with MODIS. It may also be noticed that in spring the MODIS SCF begins to fall slightly earlier compared with AVHRR and EBFM, while the increase in SCF at the end of the summer is somewhat misleading due to the different periods of coverage of the dataset, with the two remote sensing datasets ending at either September 30 (AVHRR) or October 31 (MODIS). The alternative snow model dataset, seNorge provides SCF from 2013 until 2019 inclusive. Here it can be seen that like AVHRR, there are large fluctuations in the land-averaged SCF during the period where SCF is at a minimum. These fluctuations can also be several tens of percent in magnitude. The curves in 2014 and 2015 display these large variations in the seNorge SCF quite clearly. Moreover, the lowest SCF reached in the seNorge dataset is some 20% lower than those exhibited by the MODIS and Sentinel-2 datasets. In general, the Sentinel-2 snow cover fraction follows closely the temporal variations of the MODIS estimates, though in 2016 and 2018 the Sentinel-2 SCF appears to fall marginally earlier than MODIS in the spring.

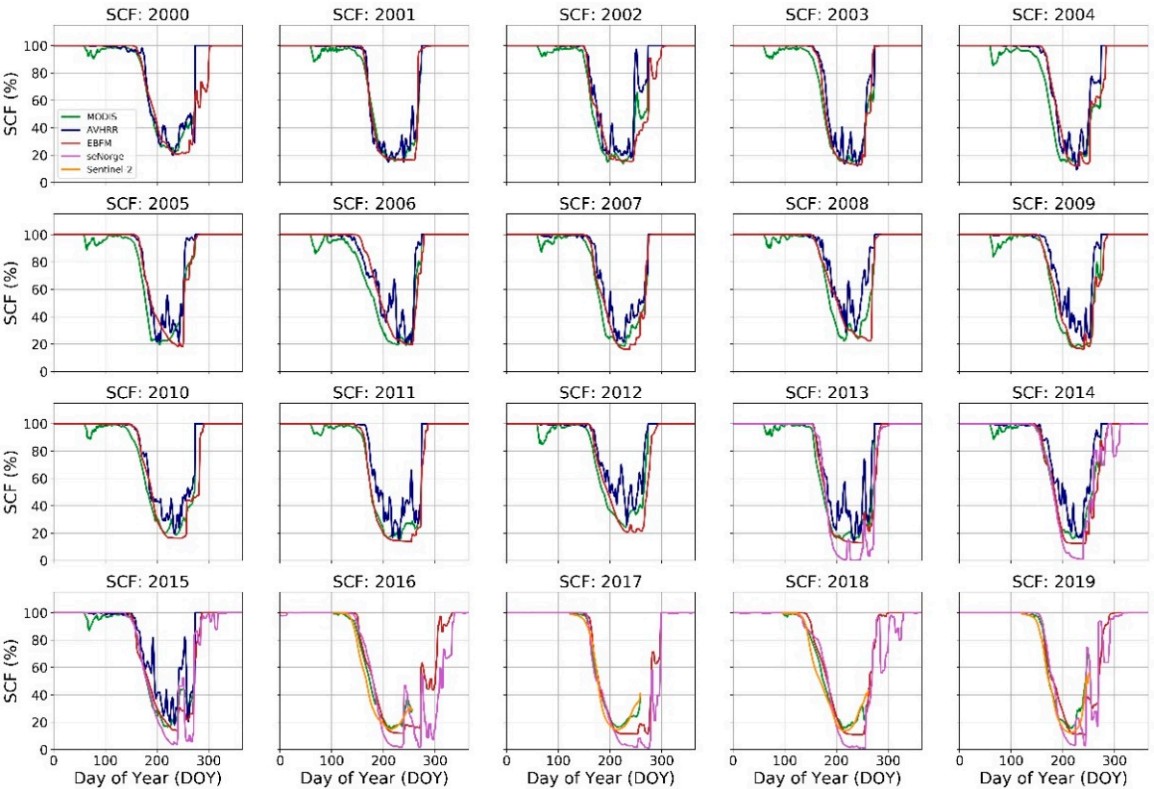

**Figure 3.** Time series comparing the land-averaged SCF derived using the MODIS dataset (green, 2000–2019), AVHRR (blue, 2000–2015), EBFM (dark red, 2000–2019), seNorge (purple, 2013–2019) and Sentinel-2 (orange, 2016–2019).

Figure 4 is a representation of the data shown in Figure 3, as a scatter plot. Here the SCF values from each dataset were plotted against the MODIS SCF, with the time series separated into two periods corresponding to data from 1 March–31 August (blue) and from 1 September–31 October (coral), or until the end of the dataset being compared. This was done to separate snow cover estimates made during the spring melt period and those

from autumn snow onset. In all but the case of the Sentinel-2 dataset, the lower resolution data from AVHRR and the snow models were plotted on the x-axis, with MODIS along the y-axis. Since the Sentinel-2 data are at higher resolution than the MODIS dataset, these were plotted along the y-axis to derive the relation that transforms the lower resolution dataset to the SCF estimates of the higher resolution sensor. Since this study endeavors to examine the differences in timing of snow disappearance between the datasets, the most critical period for dataset correction is the snow melt period. As such, the function used to transform the datasets is obtained by fitting a cubic spline function to the pairs of data obtained only in the period from 1 March–31 August (blue datapoints) to derive the general relationship with the MODIS values; this is displayed by the light blue curves. The number of scatter points in each plot reflects the size of the dataset, with the AVHRR and EBFM datasets being the largest with respectively 16 and 20 full years of data that overlap with MODIS. Of all four datasets being compared against MODIS, there is poorest agreement with the AVHRR dataset, in terms of magnitude. This is especially true when the land-averaged MODIS SCF lies the range 30–60%, with the corresponding AVHRR SCF being on average 20% greater than MODIS. On the other hand, the land-averaged SCF obtained from Sentinel-2 generally agrees well with MODIS at low (<25%) and high (>90%) snow cover fractions. At all other snow cover fractions, the Sentinel-2 snow cover fraction is in general lower than MODIS and can reach up to nearly 10% lower than MODIS. For the seNorge dataset there is generally a good but non-linear correlation with the MODIS values, when considering only the SCF data from between 1 March–31 August. There is clearly a large spread in values for SCF obtained in the period 1 September–31October, but for the melt period of interest the land-averaged SCF derived from seNorge is of the order 5–10% greater than MODIS when MODIS SCF is >30%. For the EBFM dataset, shown in Figure 4d the relationship between the MODIS SCF and the model-derived SCF is similar to that of seNorge snow model, but the EBFM estimates can be on average up to 15% greater than MODIS for the period between 1 March–31 August, as indicated by the largest offset between the fitted spline (light blue curve) and the equality line (dark blue, dashed) while EBFM estimates of SCF obtained from after September 1 which corresponds to the start of the hydrological year, are consistently lower than MODIS.

### 3.2. Geographical and Elevation Differences

The difference in annual number of days with snow cover derived from each of the data products and that obtained from MODIS was mapped. To make this geographical comparison, a binary snow map was first obtained. For the fractional snow cover products SCF was thresholded at 50%, where SCF below the threshold is defined as "no snow" and SCF greater than the threshold, "snow". Hence, for each pixel in the grid, the number of days the pixel was classified as snow covered/not-snow covered during each year of data coverage using the datasets was calculated. The difference in number of days with snow cover between AVHRR/Sentinel-2 and MODIS, and the two snow model datasets and MODIS was then calculated. For the remote sensing datasets, the difference in number of days with snow applies only to the part of the year outside of the polar night period. For the AVHRR dataset, this is the difference in number of days with snow between 1 March and 30 September, while for Sentinel 2 it is restricted to only the period 15 April–15 September. Since the two snow models produce SWE and SCF for the entire year, the full period of MODIS coverage from 1 March–31 October is used in the comparisons.

In Figure 5, the mean difference in number of days with snow derived for each of the datasets is shown. The mean difference was calculated per pixel by averaging the difference in number of days with snow over all the years in each dataset. Since the difference is calculated by subtracting the MODIS number of days with snow from the AVHRR number of days with snow, positive numbers indicate a greater number of days with snow per year on average with respect to MODIS and negative numbers indicate fewer days with snow per year on average, compared with MODIS. In the case of the AVHRR dataset, it has already been shown in Figure 4a that snow cover fraction is on average for the study

area, always greater than that estimated using the MODIS instrument, for the period of data cover 2000–2015. Figure 5a illustrates this pattern geographically, where there are large areas of blue (positive difference in number of days with snow) that correspond to low-lying valleys. Areas with light yellow tone indicate where the number of days with snow each year estimated by AVHRR and MODIS are roughly the same (zero difference). Since the AVHRR data were georeferenced to the MODIS grid, the downscaling from 4 km to 500 m pixel spacing is also clear from the square-like edges of blue areas. Areas where the difference in number of days with snow cover estimated by AVHRR was less than MODIS, indicated by the dark red areas, can also very likely be attributed to the resolution differences.

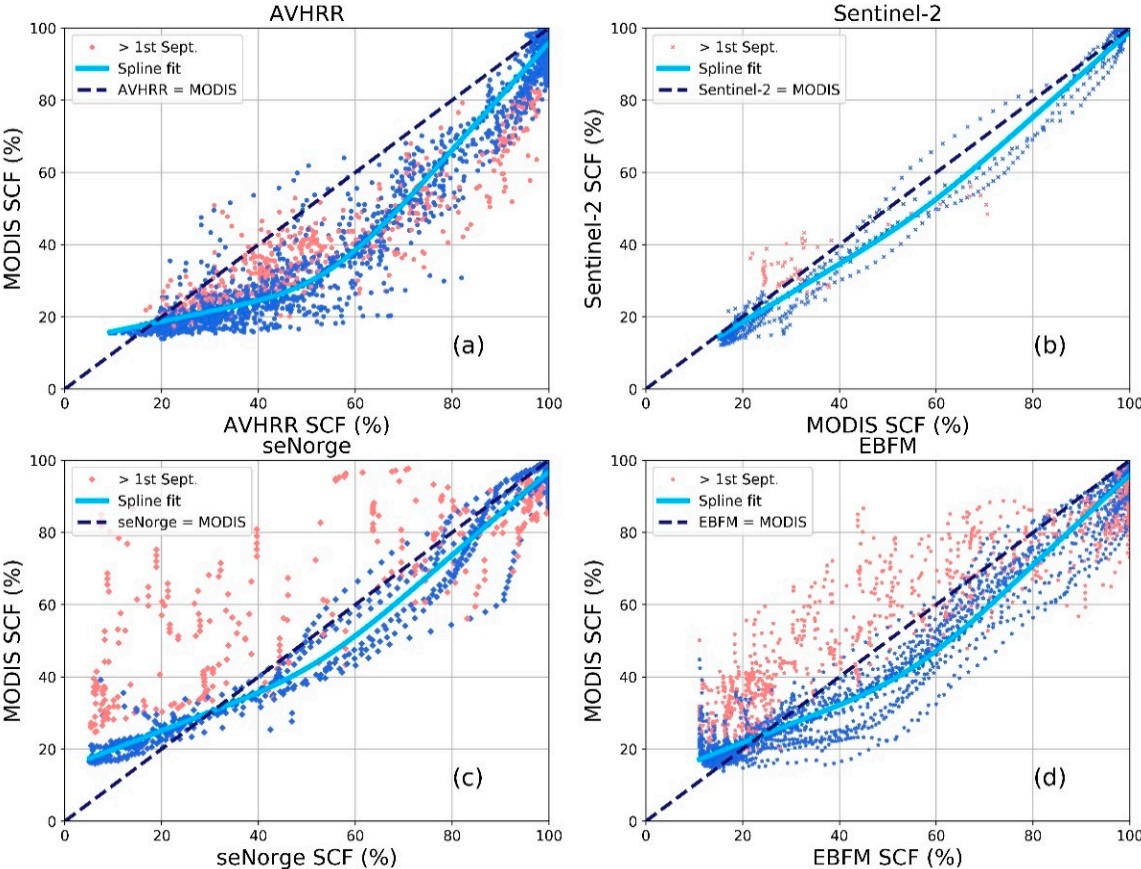

**Figure 4.** Scatter plots illustrating the relationship between the land-averaged SCF derived from MODIS and (**a**) AVHRR (2000–2015) (**b**) Sentinel-2 (2016–2019) (**c**) seNorge snow model (2013–2019) (**d**) University of Uppsala EBFM (2000–2019). In each case the datasets have been georeferenced to the MODIS 500 m grid and all days with overlapping data coverage have been used to produce the scatter plots. The time series of SCF for each dataset were split into two time periods, with blue datapoints corresponding to 1st March–31st August and coral-colored datapoints corresponding to data produced for the period 1st September–31st October.

For Sentinel-2 data on the other hand, which have much higher spatial resolution than MODIS, the difference in mean number of days with snow for the period 2016–2019, shown in Figure 5b is close to zero or below zero across the area of study, as exhibited by the prevalence of yellow and orange. This implies that MODIS always estimates a greater number of days with snow per year than Sentinel-2, with greatest differences on mountain slopes. Figure 5b would suggest that MODIS estimates greater than 60 days more with snow in these areas, compared to Sentinel-2.

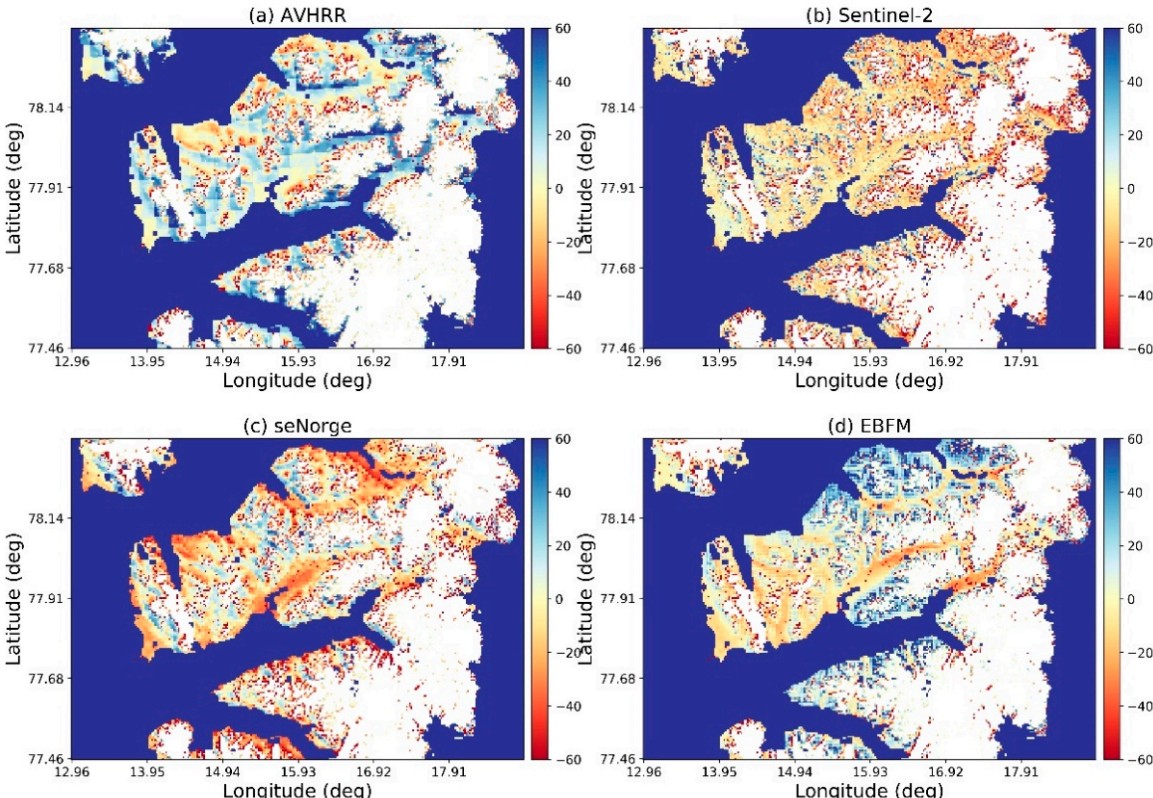

**Figure 5.** The geographical differences in the average number of days with snow per year, comparing the average number of days of snow per pixel derived using the MODIS dataset and (**a**) AVHRR (2000–2015) (**b**) Sentinel-2 (2016–2019) (**c**) seNorge snow model (2013–2019) and (**d**) University of Uppsala EBFM (2000–2019). The average number of days with snow is calculated using different number of years due to the time period of data coverage/availability as stated in the parentheses.

The comparison of mean days with snow cover between MODIS and the two models, seNorge and EBFM is shown in Figure 5c,d respectively. The geographical distribution and magnitude of the differences are quite different; for the seNorge snow cover area dataset, the snow model estimates on average fewer days with snow cover per year in the low-lying valleys and around all coastal areas, when compared with MODIS. On the other hand, blue areas corresponding to the highest elevation mountain zones indicate that the seNorge estimates on average a greater number of days with snow in these areas compared to the MODIS dataset. For the EBFM dataset shown in Figure 5d, there is also a tendency toward moderate to large underestimation in mean number of days with snow cover per year for the valley areas when compared with MODIS, as shown by the light yellow, orange and red regions. However, different to the seNorge dataset, EBFM tends to produce larger underestimation in number of days with snow in the inland parts of the valleys, whereas the largest underestimations for the valley areas in the seNorge dataset tends to be situated closer to the coastal areas. In addition, the EBFM dataset also transitions to greater number of days with snow cover compared to MODIS in the mountainous regions in the northern and eastern part of the study area but begins at much lower elevations than for the seNorge dataset. This is especially noticeable for the mountain slopes that are also located along coastal areas.

These elevation-dependent differences in the mean number of days with snow per year, between the different remote sensing and snow model datasets, and MODIS, is illustrated in Figure 6. This figure was produced by averaging the mean differences in number of days with snow (Figure 5) over all elevation zones at intervals of 200 m between 0 and 1200 m.a.s.l. using a high resolution (20 m) Digital Elevation Model (DEM). Here, the x-axis of Figure 6 represents the middle point of each elevation interval. Figure 6 therefore

demonstrates that the degree of overestimation in mean number of days with snow by AVHRR in the low altitude areas (0–200 m.a.s.l.) is in fact only of the order of 10 days on average for the whole area of interest. The magnitude of underestimation in mean number of days with snow estimated by AVHRR with respect to MODIS increases with elevation, with the data suggesting underestimations of approximately 40 days or more for all altitudes intervals above 600 m.a.s.l. As shown in Figure 5b, Sentinel-2 data exhibit of the order 10–15 fewer days with snow on average compared to MODIS at all elevations, though at the lowest altitudes the difference is only 5 days.

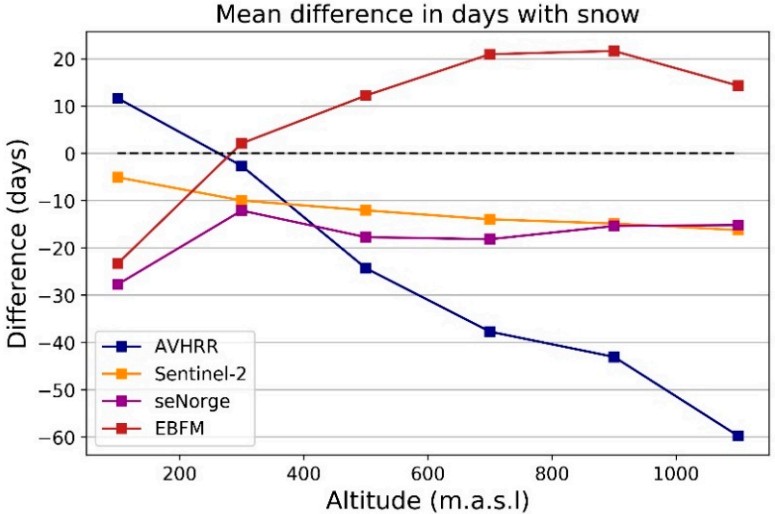

**Figure 6.** The altitude distribution of the average difference in number of days with snow for the four datasets being compared with MODIS. This figure is the equivalent altitude distribution of the data shown in Figure 5a–d.

Common to both snow models, there is a relatively large underestimation in mean number of days with snow at elevations between 0–200 m.a.s.l. when compared with the MODIS dataset. The mean number of days with snow is on average of the order of 25 days less than MODIS in this elevation zone for the EBFM and seNorge datasets. However, the elevation distributions are noticeably different for the two snow models at elevations above 400 m.a.s.l. While the seNorge dataset exhibits a negative difference in mean number of days with snow compared with MODIS on average (15–20 days) for the whole region at all elevations >200 m.a.s.l., EBFM estimates on average 10–20 days greater snow cover per year compared with MODIS when averaged over all the elevation intervals above 400 m.a.s.l., with largest differences present at elevations between 600–1000 m.a.s.l. This pattern was also described earlier for the geographical distribution of the differences shown in Figure 5d.

### 3.3. Correction of the Datasets

The Sentinel-2 dataset, with the highest spatial resolution of all the datasets but relatively small temporal period of coverage, was used to adjust the lower-resolution MODIS SCF time series by applying the obtained spline fit (Figure 4b) to the MODIS time series. With the adjusted MODIS time series, spline fits to the three remaining datasets were subsequently updated and applied to the respective time series to obtain a final corrected SCF for the three remaining lower resolution datasets (AVHRR, seNorge and EBFM). The objective of this procedure was to normalize the snow cover observations from each dataset to a baseline in order to achieve better consistency between the products. The final corrected SCF time series and the corresponding spline fits associated with these are presented in Figure 7a–d. While there remains a degree of spread in the land-averaged SCF values, the corrected time series are on average much closer to the MODIS values, as

exhibited by the final spline fits which lie close to the equality line (dark blue, dashed). In particular, the relationship between the corrected datapoints and the MODIS datasets for the melt period between 1 March–31 August is more linear compared with their original time series. This is true for all datasets.

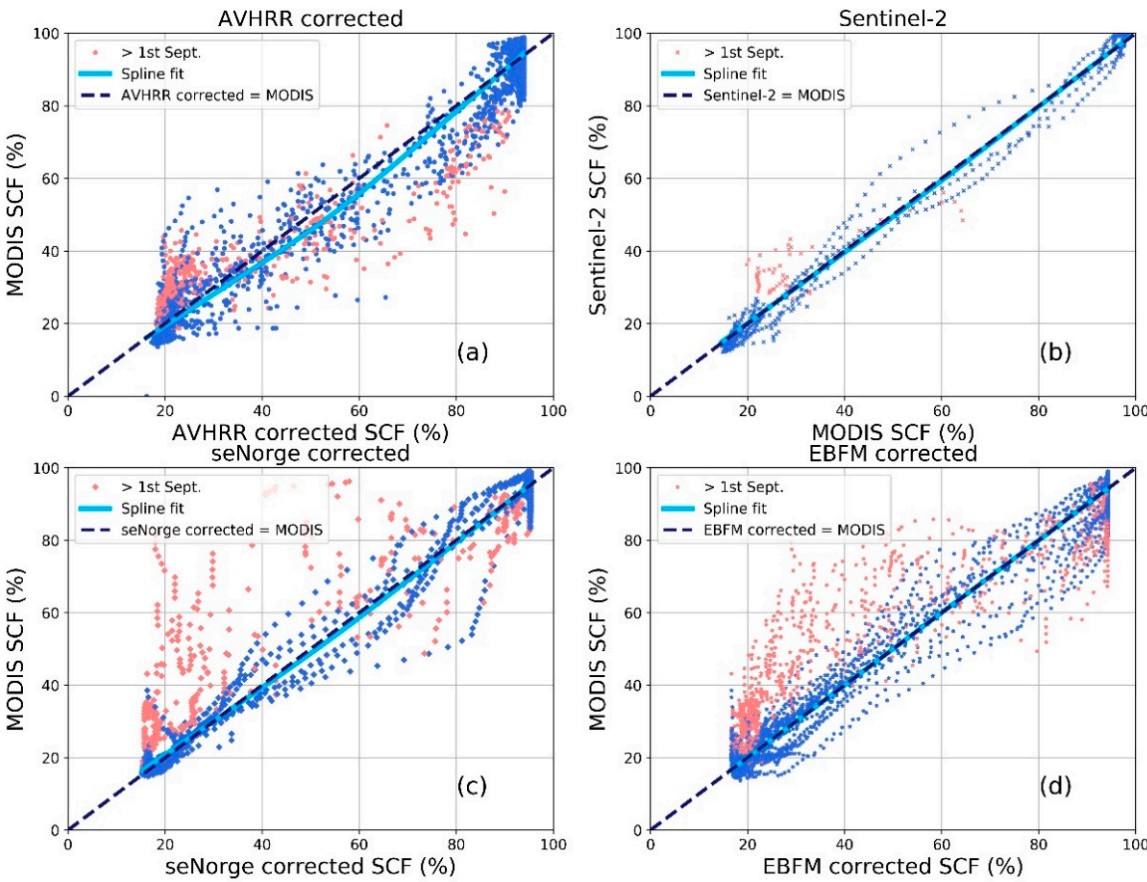

**Figure 7.** Scatter plots showing the relationship between the corrected MODIS dataset and the adjusted lower resolution datasets. The initial correction of the MODIS dataset was performed using the spline fit shown in Figure 4b. The corrected MODIS dataset was then used to obtain updated spline fits with the AVHRR, seNorge and EBFM datasets (not shown) which were subsequently applied to produce the final adjusted datasets shown in (**a**,**c**,**d**).

### 3.4. Estimation of First Snow-Free Day (FSFD)

The SCF time series produced using the different datasets were used to extract estimates of the first snow-free day, which is taken to be the point at which the SCF first falls below 50%. FSFD was estimated using the uncorrected and corrected land-averaged SCF time series for each dataset and compared with the FSFD values produced using the corresponding MODIS time series. These are shown in Figure 8a,b. Qualitatively speaking, the FSFD estimates made using the uncorrected datasets lie much further from the MODIS estimates (Figure 8a) compared with those made using the corrected SCF time series, as shown in Figure 8b. Significant improvements in the estimates of FSFD occur following correction of the SCF time series, which before the dataset was corrected, were up to 15 days greater than the corresponding MODIS FSFD estimates. Following correction of all datasets, FSFD estimates obtained from AVHRR, seNorge and EBFM all lie close to the MODIS FSFD estimates, with the corrected FSFD occurring within approximately 5 days later or earlier than MODIS. This is as expected since the spline fit made to the corrected datasets, shown in Figure 6, all closely follow the line which indicates where the two datasets would be equal.

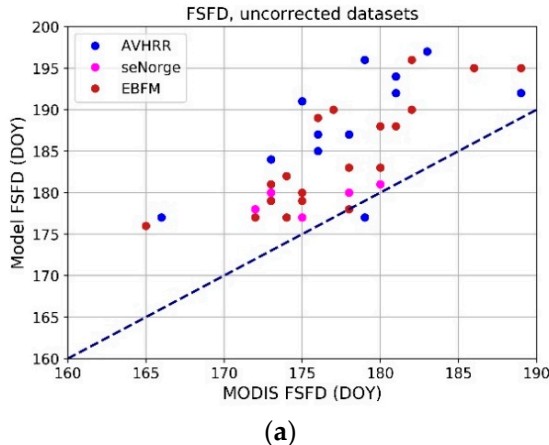
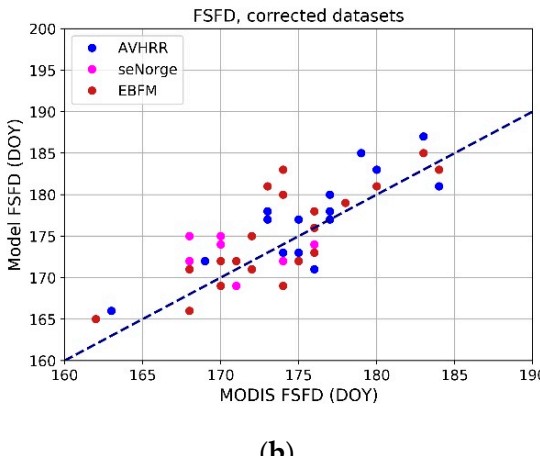

**Figure 8.** (**a**) First snow-free day estimates shown for the three uncorrected lower resolution AVHRR, seNorge and EBFM datasets, plotted against the uncorrected MODIS land-averaged FSFD estimates. A dark blue dashed line indicates where the FSFD estimates would be equal and (**b**) FSFD estimates obtained from the corrected MODIS, AVHRR, seNorge and EBFM datasets.

In the case of the AVHRR dataset, the entire 34-year time series of snow cover extent maps for 1982–2015 are used to first extract the land-averaged SCF which is then corrected using the updated spline fits. By estimating FSFD for the entire 34–year period, decadal trends in FSFD have also been calculated using both the original uncorrected AVHRR SCF time series. This has allowed us to examine differences in FSFD trends when the data had not been upscaled using the fitted spline function. For the two corrected SCF time series corresponding to the snow models, only estimates of FSFD were made and compared with MODIS, since the temporal period of coverage of each dataset was not long enough to obtain meaningful estimates of decadal trends.

For the FSFD estimates extracted using the original AVHRR time series from 1982–2015, shown in Figure 9, FSFD occurs later compared with those extracted from the corrected time series. The offset between the two linear trend lines is approximately 15 days, indicating that using the original AVHRR SCF time series results in FSFD estimates that are later by on average 2 weeks when compared with the FSFD estimates extracted from the corrected time series. Moreover, the slope of the linear trend line is also smaller, with a decadal trend in FSFD of $-3.36$ days/decade ($p = 0.08$) for the uncorrected time series, while the decadal trend in FSFD estimated with the corrected AVHRR SCF time series is $-2.81$ days/decade ($p = 0.01$). Hence, not only does the decadal trend in FSFD derived from the corrected time series become significant at the 95% confidence level, but the advance in FSFD is >0.5 days/decade slower than that suggested by the original AVHRR dataset.

### 3.5. Evaluation Metrics

To evaluate the effect of the corrections on each dataset, four metrics were selected for evaluation using both the uncorrected and corrected datasets, following the approach of recent similar studies that compare snow cover retrievals from lower and higher resolution datasets [17,30,31]. We have first calculated the mean error, which indicates the bias of the dataset being evaluated. This was done firstly using Sentinel-2 as a baseline for the MODIS snow cover dataset and subsequently for the AVHRR, seNorge and EBFM datasets with MODIS as the baseline dataset. We have therefore implicitly assumed that Sentinel-2 is more accurate than MODIS and that MODIS is more accurate than AVHRR and the two snow models for the purpose of this evaluation. The root mean-squared error (RMSE) was also calculated, as well as the Spearman rank correlation coefficient. The Spearman rank correlation coefficient was chosen over the Pearson correlation coefficient since it is known to be more appropriate for non-linear relationships between datasets. For both the uncorrected and corrected datasets, each of the four metrics was calculated for two

cases: firstly, using the period from 1 March–31 August and second, the period from 1 September–31 October as a basis for the calculations.

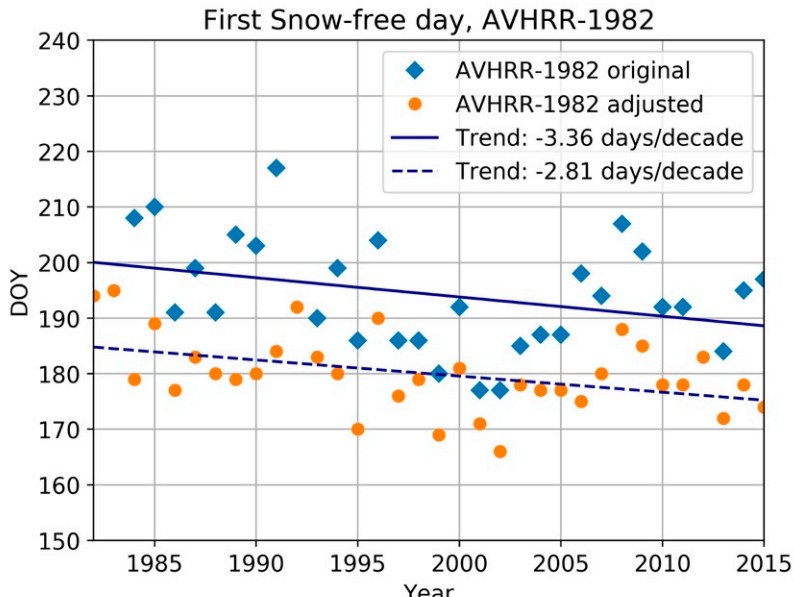

**Figure 9.** Estimates of FSFD obtained using the original (blue diamonds) 34-year AVHRR time series of the land-averaged SCF for Nordenskiöld Land, from 1982–2015 and the adjusted (orange circles) AVHRR SCF time series. Linear fits were made to each time series, shown by the dark blue solid line for the original AVHRR datasets and the dashed line for FSFD obtained using the adjusted dataset. Decadal trends derived from the linear fits are indicated in the legend.

Table 2 summarizes the RMSE, mean error (bias) and Spearman correlation coefficient for the AVHRR, seNorge and EBFM datasets for the main period of interest (1 March –31 August) as well as the autumn data acquired after 1 September, which are stated in italic. The upper section presents the metrics calculated using the uncorrected SCF time series, while the lower section of Table 2 displays the same metrics calculated following correction of the datasets. Sentinel-2 was used as a baseline for obtaining the spline model with which corrections to the MODIS time series was made and we show only the metrics that were calculated for the datasets sharing the same (MODIS) baseline.

**Table 2.** Evaluation metrics for the uncorrected and corrected (land-averaged) SCF time series in the period 1 March–31 August. Values in parentheses correspond to the metrics calculated only for data acquired in the period 1 September–31 October (or 30 September for AVHRR).

| Metric (*uncorrected*) | AVHRR | seNorge | EBFM |
|:---:|:---:|:---:|:---:|
| RMSE (%) | 11.22 (*17.07*)) | 6.82 (*23.84*) | 7.45 (*13.86*) |
| Mean error (%) | 7.85 (*13.56*) | 2.11 (*–12.05*) | 3.65 (*–3.66*) |
| R | 0.85 (*0.89*) | 0.85 (*0.76*) | 0.91 (*0.94*) |
| (*corrected*) | | | |
| RMSE (%) | 5.95 | 4.64 | 4.80 |
| Mean error (%) | 0.39 | −0.19 | 0.0 |
| R | 0.87 | 0.90 | 0.91 |

Firstly, comparison of the metrics calculated for the spring (1 March–31 August) and autumn (1 September–31 October) data reinforces the patterns described by the scatter plots shown in Figure 3. For all three datasets, there is greater spread in the data acquired

after 1 September, as indicated by the larger RMSE values; the mean error is also greater and in the case of the snow models, of the opposite polarity compared with the spring data. Comparison of the metrics calculated before and after corrections were made to the time series shows an obvious improvement and reduction in the RMSE and mean error. Greatest changes in RMSE occur in the correction of the AVHRR time series, resulting in RMSE being nearly halved, from 11.22 to 5.95%. The mean error is also reduced from 7.85% to 0.39%, indicating that the positive bias, or overestimate becomes almost minimal following correction of the dataset. This is indeed reflected by the scatter plot in Figure 6a. The Spearman correlation coefficient was only marginally increased by correcting the time series and this is not surprising since there was qualitatively a good and non-linear correlation between AVHRR and MODIS before corrections were made. For the MODIS dataset itself, the corrections made to the time series using the relationship with Sentinel-2 also resulted in a small reduction in RMSE (not shown), as well as removal of the positive bias of 2.8% which was present in the uncorrected time series. The Spearman correlation coefficient of 0.97 remained unchanged. For the two snow models, the corrections made using the spline fits also resulted in a reduction in the RMSE by between 2–2.5% while the slight positive biases in both uncorrected datasets were reduced to almost zero. For the seNorge snow model, the Spearman correlation coefficient increased from 0.85 to 0.90 while for the EBFM dataset there was no change.

## 4. Discussion

This study has utilized snow cover observations from three optical remote sensing satellites and two snow models and made comparisons of the land-averaged snow cover fraction and the derived FSFD over Nordenskiöld Land in central Svalbard in the period 2000–2019. In this section we review and discuss the results presented in Section 3.

### 4.1. Comparison of the Datasets

A consistent pattern was identified, whereby lower resolution datasets are found to overestimate SCF when compared with a higher spatial resolution dataset. SCF estimated using the AVHRR dataset were found to be up to some 20% greater than the corresponding MODIS estimates, especially for intermediate snow cover fractions in the range 50–60%. Similar findings have earlier been reported using AVHRR observations over the Canadian Arctic, where an evaluation of the NOAA AVHRR snow cover dataset was shown to consistently overestimate snow cover extent during spring snowmelt period [32], ultimately resulting in apparent delays of up to 4 weeks in estimates of melt onset. We also find that SCF overestimations in the AVHRR dataset are predominantly found at lower elevation areas, but with a tendency to underestimate the number of days with snow in higher elevation mountainous areas when compared with the MODIS dataset. Since the topography in mountainous areas can change over spatial distances smaller than the 4 km resolution of the AVHRR data, the observations would indicate that the AVHRR instrument is unable to capture the spatial variations in snow cover occurring over distances smaller than the AVHRR resolution in these areas of high relief, resulting in an overall underestimation with respect to higher resolution datasets. However, since the land-averaged snow fraction in the AVHRR is systemically greater than MODIS, the findings would suggest that the underestimation in snow cover at higher elevations does not contribute greatly to the overall differences between MODIS and AVHRR.

Similarly, it was also found that the MODIS SCF was on average 5–10% greater than SCF estimates made by the Sentinel-2 sensor, dependent on the magnitude of the snow cover fraction. It has recently been demonstrated that for high resolution sensors such as Sentinel-2, both thresholding and regression-based algorithms can systematically overestimate fractional snow cover when the data are aggregated to lower resolution grids [17], such as that of the MODIS dataset which was done in this work. This suggests that the Sentinel-2 estimates of the land-averaged SCF could in fact be lower than was illustrated, had more superior retrieval methods such as spectral unmixing been implemented to

estimate SCF. As a result, the difference between the Sentinel-2 and MODIS SCF estimates would be greater than what has been shown in this study. An additional factor which has not been accounted for in this work is the effect of terrain shadow on Sentinel-2 NDSI. In autumn, the low sun elevation on Svalbard causes topographic shadow, which could be misinterpreted as snow cover in the NDSI values, which would also cause the derived Sentinel-2 snow fraction to be greater than if these effects were corrected for. A systematic study of the impact of such shadow for all the remote sensing datasets is needed to quantify this problem. Other recent studies that have investigated methods to retrieve fractional snow cover from Sentinel-2 have modelled the NDSI-snow cover fraction relation as a non-linear sigmoid-shaped function [31], which may present a better approach for calibrating the MODIS data against Sentinel-2 in future. However, since this study was based on a test site in alpine mountainous terrain, the calibration coefficient may not necessarily be universal and may need to be revised for polar regions such as Svalbard and is the main reason this approach was not adopted in this study.

For the comparison between MODIS and snow cover fraction derived from the two snow model datasets that were produced at 1 km resolution, there was also an offset of 5-15% between the MODIS dataset and SCF provided by the seNorge snow cover product or by thresholding the EBFM SWE dataset, with slightly closer agreement for the seNorge dataset when only data from the spring melt period, between 1 March–31 August were considered. In contrast to the AVHRR and Sentinel-2 comparisons, both snow models exhibited distinct differences in the relationship with MODIS SCF when data from the spring melt period and autumn snow onset (September 1 onwards) were compared. For the seNorge model, there was a consistent underestimation of SCF compared with MODIS for the post-September 1 data, while SCF estimated by the snow models during the spring melt period from 1 March–31 August, were almost always greater than the corresponding MODIS SCF. This pattern was not observed in the AVHRR and Sentinel-2 datasets, where SCF estimated in the period after 1 September tended to follow the same relationship with MODIS as those which had been obtained from the spring melt period. Furthermore, the snow models tended to underestimate SCF for low SCF (<20%), which is true of the data from both the spring melt period (1 March–31 August) as well as from after 1 September, with the geographical variations showing that the underestimations were greatest in lower elevation zones, as indicated in Figures 5 and 6. During the spring snowmelt period, this may for example be attributed to snow accumulation in terrain formations such as couloirs and lee slopes which melt later. These may also contribute to perennial/multi-year snow patches that are present during the summer months and can be captured by MODIS but not the snow models. On the other hand, since the calculation of the average number of days with snow includes data from part of the autumn period, the underestimation of snow is due to a combination of the models missing snow during spring melt as well as underestimating new snow in the autumn, which may have other underlying reasons.

There were also elevation-dependent differences in snow cover obtained from the two snow models. For SCF derived from the EBFM SWE dataset, there was on average more days with snow compared with MODIS at altitudes above 400 m.a.s.l. In contrast, the seNorge snow model exhibited consistently fewer days with snow on average compared with MODIS, at all altitudes. However, there are differences between the two snow model data products as well as the method by which they are derived. The EBFM dataset covers a period of 20 years between 2000–2019, while the seNorge product only covers 7 full years in the latter part of the same period, from 2013–2019. On the other hand, we have also analyzed the EBFM data only for the 2013–2019 interval, but only minor differences were found, and these are shown in Figure 10; there is still the same pattern of overestimation in the land-averaged SCF compared with MODIS in the 1 March–31 August period and these were predominantly found at higher elevations. The underestimation in average number of days with snow was still confined to the lower elevation valley areas but were in fact more pronounced than when averaged over the 20-year period. On the other hand, both

the RMSE and mean error in the EBFM SCF estimates were reduced to 5.63% and 1.91% respectively, which are slightly lower than for the 2000–2019 period, given in Table 2.

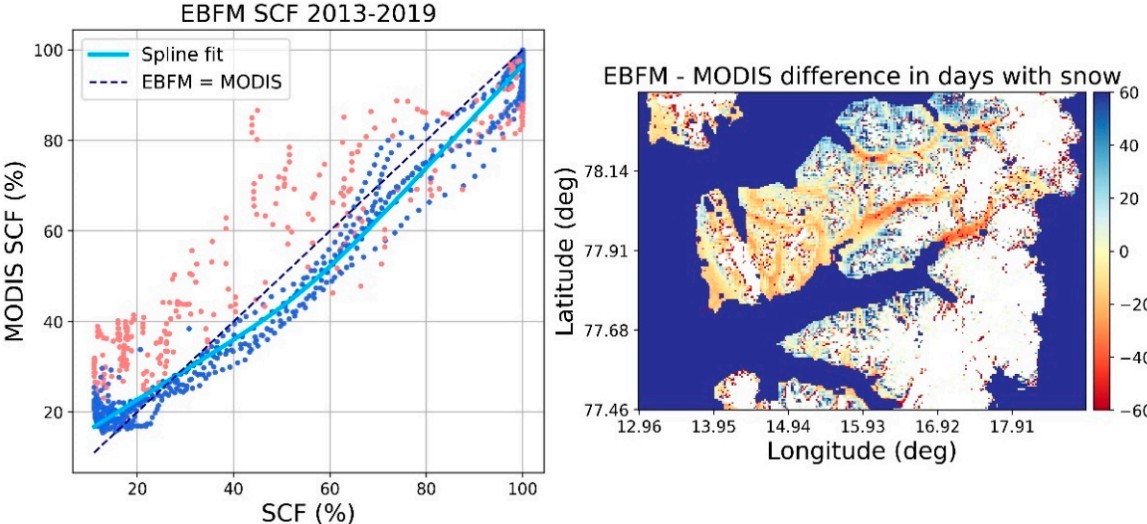

**Figure 10.** (**left**) Scatter plot showing the relationship between MODIS and the EBFM land-averaged snow cover fraction using data corresponding to the seNorge period (2013–2019) and (**right**) the average difference in number of days with snow comparing EBFM and MODIS for 2013–2019.

Moreover, the seNorge dataset is produced using the AROME NWP model forecast data and model archives for the historical period as inputs, while for EBFM the input is the HIRLAM reanalysis data. Thus, the snow models are not driven by the same inputs and cannot be expected to produce outputs that behave identically. Since there was little change between the geographical differences in average number of days with snow for the EBFM dataset shown in Figures 5d and 10, the dissimilarities between the seNorge and EBFM models are most likely attributed to differences in the way the models distribute new or melting snow, as well as due to different meteorological input data. In addition, the land-averaged SCF has been estimated from the EBFM SWE product by first converting the SWE maps into a binary snow cover map using a fixed threshold of 0.003 m. The choice of SWE threshold may therefore have contributed to the apparent greater number of days with snow in higher elevation zones in the EBFM snow cover dataset for certain years where the optimum threshold was found to be greater. However, we have chosen to use a fixed value based on the threshold which was found to produce best agreement with MODIS for the majority of the EBFM dataset, thereby maintaining a consistent method of snow cover extent extraction from the SWE dataset.

### 4.2. Correction of the Datasets

Using the spline fit to the MODIS SCF time series and the higher resolution Sentinel-2 dataset, the MODIS dataset was adjusted and subsequently utilized to obtain updated spline fits with the three remaining datasets. In doing so, a correction to the lower resolution SCF time series from the AVHRR instrument and the two snow models was also made. Evaluation metrics calculated for the MODIS time series in this study, using Sentinel-2 as a baseline before correction indicated a mean error, or bias of approximately 5.7%. A similar study that used high-resolution (0.5 m) terrestrial photography as a reference for the evaluation of Sentinel-2, Landsat 8 and MODIS datasets for a case study area in north-western Svalbard [17], reported comparable values, with the MODIS MOD10A1-v6 dataset exhibiting a mean error of 5% with respect to the high-resolution reference dataset. The RMSE of 8.6% in the uncorrected dataset is somewhat smaller than the RMSE obtained in their study, which was of the order of 14–15%, but here we have used Sentinel-2 data rather than very high-resolution optical images as a reference. In several earlier studies, the RMSE

of the dataset being evaluated is often reduced when the reference data are aggregated to coarser resolution [17,31] as was done with the Sentinel-2 data in this study. Moreover, at the resolution of the MODIS data, the correction using the spline relationship resulted in just 1% reduction in RMSE and no change in correlation, while the mean error was reduced to 0%. Similarly, we found that largest decreases in the metrics being evaluated, occurred in the mean error following correction of the AVHRR, seNorge and EBFM datasets, while more modest reductions were observed in RMSE. The exception to this is for the AVHRR dataset, where RMSE was almost halved as a result of the corrections. The Spearman correlation coefficient remained virtually unchanged for AVHRR and EBFM while an increase in correlation from 0.85 to 0.90 occurred from the correction to the seNorge dataset. Hence it may be deduced that the main effect of the SCF time series using a spline model fit, was to almost remove the bias in the datasets.

A pattern of earlier FSFD following correction of the AVHRR SCF time series was also found. In this case FSFD obtained with the uncorrected AVHRR SCF time series was on average 15 days later than FSFD estimated using the corrected AVHRR SCF time series. In both cases, decadal trends revealed that FSFD became earlier over the 34-year period, but the advance in FSFD estimated using the uncorrected SCF dataset, of $-3.36$ days/decade ($p > 0.05$) was greater than the advance observed from the corrected dataset, which was found to be $-2.81$ days/decade ($p < 0.05$), by 0.5 days per decade. Hence, the results obtained in this study would imply that not only does the lower resolution AVHRR snow cover data tend to overestimate snow cover with respect to higher resolution datasets, but it also results in estimates of timing of snow disappearance that can be too late by up to two weeks, as well as a small overestimation in the decadal advance in FSFD, or the rate at which snow cover is declining. This result can have significant impacts, for example in applications where remote sensing observations of snow cover are used as inputs to calibrate climate and hydrological models. In such cases the accuracy of predictions for both present and future climate changes will be dependent on the representation of snow processes, and there is therefore a need to ensure that these parameters are reliable.

On the other hand, the calibration of the AVHRR data are based only on the 15 years of overlap with MODIS and may not necessarily be valid for the earlier part of the AVHRR dataset (1982–1999) which was not used in the comparisons. However, the decadal trends in FSFD found in this work are comparable to those published in a recent where the same 34-year AVHRR snow cover dataset was used to identify the relationship between snow cover variability and sea ice variability [33]. In this work, the authors found a decadal trend of $-2.6$ days/decade change in melt onset, defined as the point where SCF crosses a threshold of 95%, rather than examining the timing of snow disappearance which uses an SCF threshold of 50% in this work. Nevertheless, it should be noted that the melt onset trend found by their study was calculated using the snow fraction averaged over the entire Svalbard archipelago, whereas this study focuses only on the Nordenskiöld Land region. Therefore, the similar trends found in both this study and that of [33] support each other, despite the differences in study area as well as the snow melt parameter analyzed.

Both uncorrected snow model datasets produced FSFD estimates that were consistently later than MODIS by up to 15 days, with largest differences exhibited by the EBFM (SWE) dataset. The seNorge dataset resulted in FSFD estimates that were later than MODIS by around 5 days, both before and after correction with the spline fit. Since it was earlier also found that the seNorge dataset estimated on average fewer days with snow per year compared with MODIS, the result that the seNorge FSFD estimates are consistently later than MODIS would indicate that the discrepancy is most likely attributed to a much later onset of snow in the autumn, producing fewer days with snow compared with MODIS. This is to some extent verified in the comparison of the yearly SCF time series shown in Figure 3, where it was observed that the seNorge SCF rises later in autumn in relation to the other datasets, as well as exhibiting a lower minimum. Datapoints corresponding to SCF estimates obtained after September 1 for the seNorge dataset also point to much lower SCF than MODIS (Figure 4c) which further verifies this conclusion. Possible

reasons for this difference may lie in the NWP input, if for example the model forecasts have a warm bias or too dry conditions, leading to too little early autumn snowfall in the 1 September–31 October period. For the EBFM dataset, the uncorrected land-averaged SCF time series also produced much later FSFD compared with MODIS, but geographically the low elevation areas exhibited on average fewer days with snow than MODIS, while higher elevation mountain slopes were found to have a greater number of days with snow on average compared with MODIS and would therefore be expected to contribute to later snow disappearance. Therefore, the temporal variations in SCF produced by EBFM may be in better agreement with MODIS at higher elevations, while being poorer at lower elevations. Moreover, there exist differences between the snow models and remotely sensed SCF for the period when snow cover is decreasing in the spring and when snow cover begins to increase again after the summer minimum snow cover extent, which suggests the need for a separate treatment in order to produce accurate corrections to the time series. The results found in this study therefore show that there is potential to integrate remote sensing observations into the calibration of snow models in order to improve the description of snow cover produced by the models.

## 5. Conclusions

Accurate maps of snow cover and characterization of the dynamic processes are critical in applications such as calibration of hydrological models and climate predictions, and especially so in regions where seasonal snow cover is responding rapidly to ongoing changes in climate. This study has investigated the similarities and differences between snow cover observations over Nordenskiöld Land in Svalbard, obtained by three optical remote sensing datasets and two snow models. The purpose of this work was to attempt to use high spatial resolution snow cover observations to make corrections to earlier, lower spatial resolution snow cover products to reconstruct long term snow cover datasets at both high spatial and temporal resolution. To achieve this, relationships between the higher and lower resolution datasets were obtained for the land-averaged snow cover fraction over the study area. Sentinel-2, with its high spatial resolution, was first utilised to adjust the moderate resolution MODIS dataset, which had excellent temporal overlap with the AVHRR dataset as well as the two snow models. This adjusted MODIS dataset was subsequently used to correct the lower resolution datasets and estimates of the timing of snow disappearance were made. For all the uncorrected datasets, estimates of FSFD were found to be later than MODIS FSFD by 10–15 days. Following correction of these datasets to the higher resolution of the MODIS dataset, FSFD estimates were significantly improved and varied by up to $\pm5$ days from the MODIS estimates. Furthermore, the decadal advance in FSFD estimated from the uncorrected 34-year AVHRR time series were found to be 0.5 days/decade greater than the decadal trend in FSFD following correction of the AVHRR time series, indicating that interpretation of lower resolution datasets requires some care when put in the context of climate-related change.

This work has demonstrated that there is potential to improve the consistency in snow cover observations made using earlier generation remote sensing instruments which have lower resolution than current day sensors that have comparatively high temporal and spatial resolution. Specifically, we have presented a method to implement relatively simple corrections to land-averaged snow cover fraction estimated by different remote sensing datasets and snow models. This approach has been shown to produce significant updates in estimates of snow disappearance timing and decadal trends in this parameter. However, since this study has focused on improving the consistency between snow cover observations at a land-averaged scale, further work is required to upscale lower resolution snow cover datasets at the pixel level and reconcile the associated geographical and elevation dependent differences in snow cover made by remote sensing and snow model datasets, as was highlighted in this study. Ultimately, it would be desirable to reproduce older snow cover maps at the high resolution of the newest sensors. This could for example be achieved by establishing a statistical average of the snow cover distribution at high

spatial resolution, given a land-averaged snow cover fraction obtained from a lower resolution dataset.

**Author Contributions:** Conceptualization, E.M.; methodology, E.M. and H.V.; software, H.V.; formal analysis, H.V.; investigation, H.V.; resources, E.M., W.J.J.v.P., V.A.P., T.S., M.A.K., S.R.K.; writing— original draft preparation, H.V.; writing—review and editing, H.V., E.M., W.J.J.v.P., V.A.P., T.S., M.A.K., S.R.K.; project administration, E.M.; funding acquisition, E.M. All authors have read and agreed to the published version of the manuscript.

**Funding:** The work was partly based on the results of the project SATMODSNOW: https://zenodo. org/record/4294072#.YKO46bUzbIU (accessed on 18 May 2021) funded by the Research Council of Norway under the project Svalbard Integrated Arctic Earth Observing System—Infrastructure development of the Norwegian node (SIOS-InfraNor Project. 801 No. 269927. WvP acknowledges funding from the Swedish National Space Agency (project 189/18).

**Data Availability Statement:** Several of the datasets used in this study are available via the SIOS data access portal: https://www.sios-svalbard.org/metadata_search (accessed on 18 May 2021). The MODIS dataset is described in Vickers et al. (2020) https://www.mdpi.com/2072-4292/12/7/1123 (accessed on 18 May 2021). For the Uppsala Snow model: https://bit.ly/3nkfu18 (accessed on 18 May 2021) and AVHRR snow cover extent maps can be accessed at https://thredds.met.no/thredds/ dodsC/arcticdata/sios/SvalSCE-agg.html (accessed on 18 May 2021). SeNorge for Svalbard is not yet currently publicly available but will be published later on www.senorge.no.

**Acknowledgments:** MODIS Terra Snow cover data were retrieved from NSCDC NASA DAAC: National Snow and Ice data Center; Copernicus Sentinel-2 data were retrieved from ESA SciHub.

**Conflicts of Interest:** The authors declare no conflict of interest.

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
