# Peer review of "A Compilation of Snow Cover Datasets for Svalbard: A Multi-Sensor, Multi-Model Study"

_remotesensing, doi:10.3390/rs13102002_

Round 1
Reviewer 1 Report
Snow cover is a crucial component in climate and hydrology systems. However, the present snow cover datasets show spatial and temporal consistency. This paper “A compilation of snow cover datasets for Svalbard: a multi- sensor, multi-model study” compared three remote sensing and two modeled snow cover datasets, analyzed their difference, and made a inter-calibration between them based on sentinel-2. The descriptions of material, methods and results are clear. However, some problems and questions should be resolved to improve the paper.
Major comments:
Section 2 and section 3 should be reorganized. Some simple methods can be described in section 3. The data processes are suggested to present using flowchart figure to avoid repeat description.
Section 4 is not logically organized and suggested to separate into two or three subsections.
sentinel 2 has higher spatial resolution, but does not mean higher accuracy. this paper focus on assessing and improving the consistence between different datasets, but not accuracy evaluation. Furthermore, the spline correction was performed based on land average, and it will not improve the snow cover distribution at high spatial resolution. So some conclusion should be revised.
Pleas see detailed comments and suggestions as annotation in manuscript.

Author Response
We are grateful for the thorough feedback from the reviewer and for the points they have made regarding our manuscript. Please find a pdf file outlining how we have revised our manuscript in response to the feedback.

Reviewer 2 Report
This manuscript made a comprehensive analysis on snow cover datasets for Svalbard.
This work is meaningful in this field. However, some other problems in the manuscript are still concerned in the following:
- The abstract could be simplified.
- The legend is missing in Figure 1.
- The organization of this manuscript should be added to the end of the introduction.
- A grammar mistake in the sentence“Despite that the study area and time period was limited in extent…”.
- The mathematical expressions should be included in“2.9 Evaluation metrics”.
- For MODIS snow cover products, they are usually contaminated by clouds, as in “DOI: 10.5194/hess-18-4579-2014”, “DOI: 10.5194/hess-23-2401-2019”, and then reconstructed by a series of methods. This point should be indicated in the text.
Author Response
We are grateful for the feedback from the reviewer and for the helpful points they have made regarding improvement of our work. Please find a pdf file outlining how we have revised our manuscript in response to the feedback.

Round 2
Reviewer 1 Report
I think authors have modified the manuscript based on my previous comments. It can be accepted for publication.